# Macrophage-released ADAMTS1 promotes muscle stem cell activation

Hongqing Du[1], Chung-Hsuan Shih[1], Michael N. Wosczyna[2], Alisa A. Mueller[2,3], Joonseok Cho[1], Abhishek Aggarwal[1], Thomas A. Rando[2,3,4] & Brian J. Feldman[1,3,5,6]

Coordinated activation of muscle stem cells (known as satellite cells) is critical for postnatal muscle growth and regeneration. The muscle stem cell niche is central for regulating the activation state of satellite cells, but the specific extracellular signals that coordinate this regulation are poorly understood. Here we show that macrophages at sites of muscle injury induce activation of satellite cells via expression of *Adamts1*. Overexpression of *Adamts1* in macrophages in vivo is sufficient to increase satellite cell activation and improve muscle regeneration in young mice. We demonstrate that NOTCH1 is a target of ADAMTS1 metalloproteinase activity, which reduces Notch signaling, leading to increased satellite cell activation. These results identify *Adamts1* as a potent extracellular regulator of satellite cell activation and have significant implications for understanding the regulation of satellite cell activity and regeneration after muscle injury.

[1] Department of Pediatrics, Stanford University, Stanford, CA 94305, USA. [2] Department of Neurology and Neurological Sciences, Stanford University, Stanford, CA 94305, USA. [3] Program in Cancer Biology, Stanford University, Stanford, CA 94305, USA. [4] Paul F. Glenn Laboratories for the Biology of Aging, Stanford University, Stanford, CA 94305, USA. [5] Stanford Cardiovascular Institute, Stanford University, Stanford, CA 94305, USA. [6] Program in Regenerative Medicine, Stanford University, Stanford, CA 94305, USA. Correspondence and requests for materials should be addressed to B.J.F. (email: feldman@stanford.edu)

The progressive activation and differentiation of satellite cells is critical for proper skeletal muscle growth and muscle regeneration after injury[1, 2]. This cascade is initiated when satellite cells are activated to break quiescence, progress through differentiation, and fuse to nascent or injured muscle fibers[2, 3]. Therefore, elucidating the signals and pathways that regulate this cascade is central to understanding muscle physiology and could provide a foundation for developing novel therapies for the treatment of muscle disorders and regenerative medicine.

Activation of satellite cells occurs in response to a variety of chemical, physical and physiological cues to mediate muscle tissue homeostasis and regeneration[4–7]. The specialized niche of satellite cells, which are located between the basal lamina and the myofiber, is a critical element in the regulation of satellite cell quiescence and activation[8–11]. For example, activated Notch signaling, which is directly regulated by proximal extracellular signals, is a well-studied example of a potent pathway that plays an important role in maintaining satellite cell quiescence[5, 6, 12]. In addition, ADAM10, an enzyme known to promote Notch signaling[13], was found to have a role in the maintenance of the quiescent state[14]. Yet, in spite of the apparent canonical role of Notch signaling in the regulation of satellite cell activation, the extracellular triggers that inhibit Notch signaling and promote satellite cells to break quiescence and differentiate are largely unknown.

Here we describe our discovery that macrophages, which are enriched at the site of muscle injuries, secrete a protein called ADAMTS1 (A Disintegrin-Like And Metalloproteinase With Thrombospondin Type 1 Motif). ADAMTS1 contains two disintegrin loops and three C-terminal thrombospondin type-1 motifs. We established that ADAMTS1 functions as an extracellular signal to satellite cells that promotes activation. We also found that constitutive overexpression of *Adamts1* in macrophages accelerates satellite cell activation and muscle regeneration in young mice. Our data indicate that the mechanism

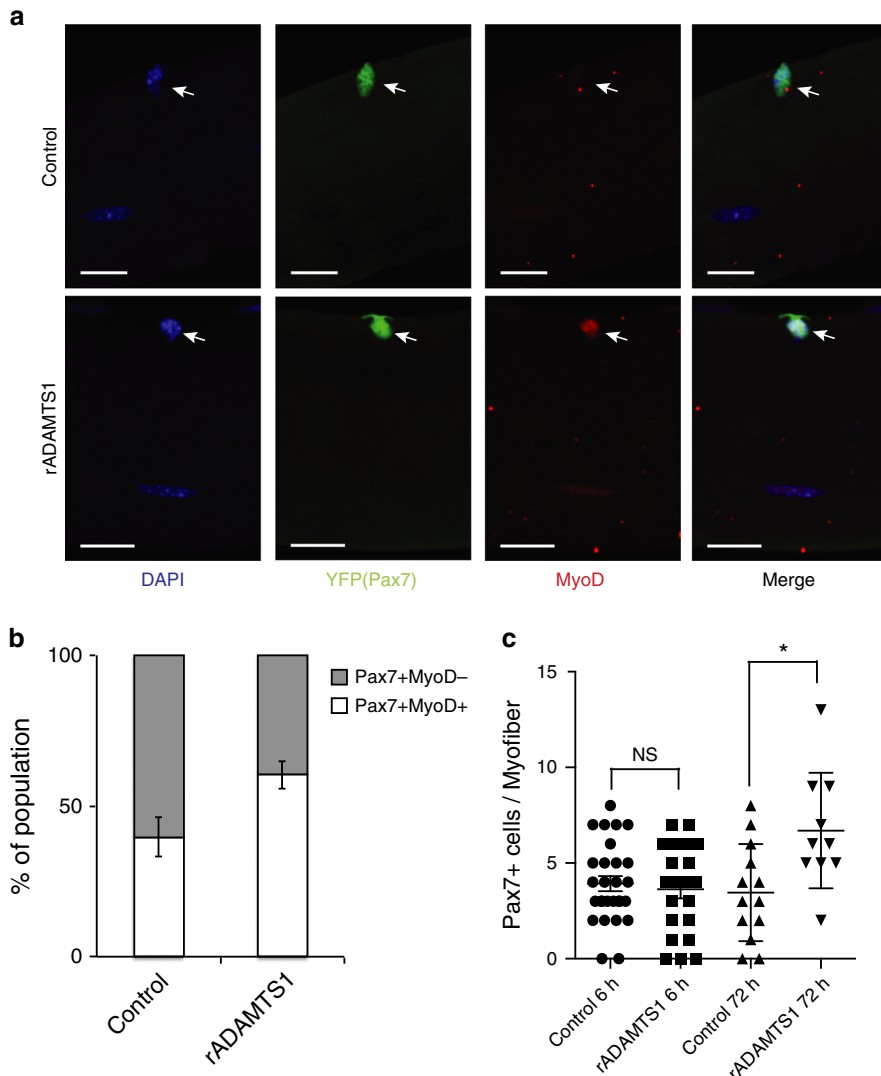

**Fig. 1** ADAMTS1 activates satellite cells. **a** Representative confocal images of myofibers with associated MyoD-negative (*upper*) and MyoD-positive (*lower*) YFP+ (Pax7+) satellite cells isolated from extensor digitorum longus (EDL) muscles of tamoxifen-treated Pax7[CreER/+];R26[eYFP/+]mice and treated with control (*upper*) or rADAMTS1 (1.4 µg/ml) (*lower*). Activated YFP+ (Pax7+) satellite cells were identified by MyoD (*red*) staining and nuclei were stained with DAPI (*blue*). *Scale bars* = 20 µm. **b** Quantification of the increased proportion of activated satellite cells on myofibers after exposure to rADAMTS1 medium compared to control (n = 75–104 myofiber-associated satellite cells per condition; n = 4 EDL replicates. The mean of the replicates is graphed. P = 0.0023. **c** Quantification of the number of YFP+ (Pax7+) satellite cells per myofiber across the conditions. Each point on the graph represents one myofiber. *P < 0.05. *Error bars* represent s.e.m. Statistical significance tested using paired *t*-tests

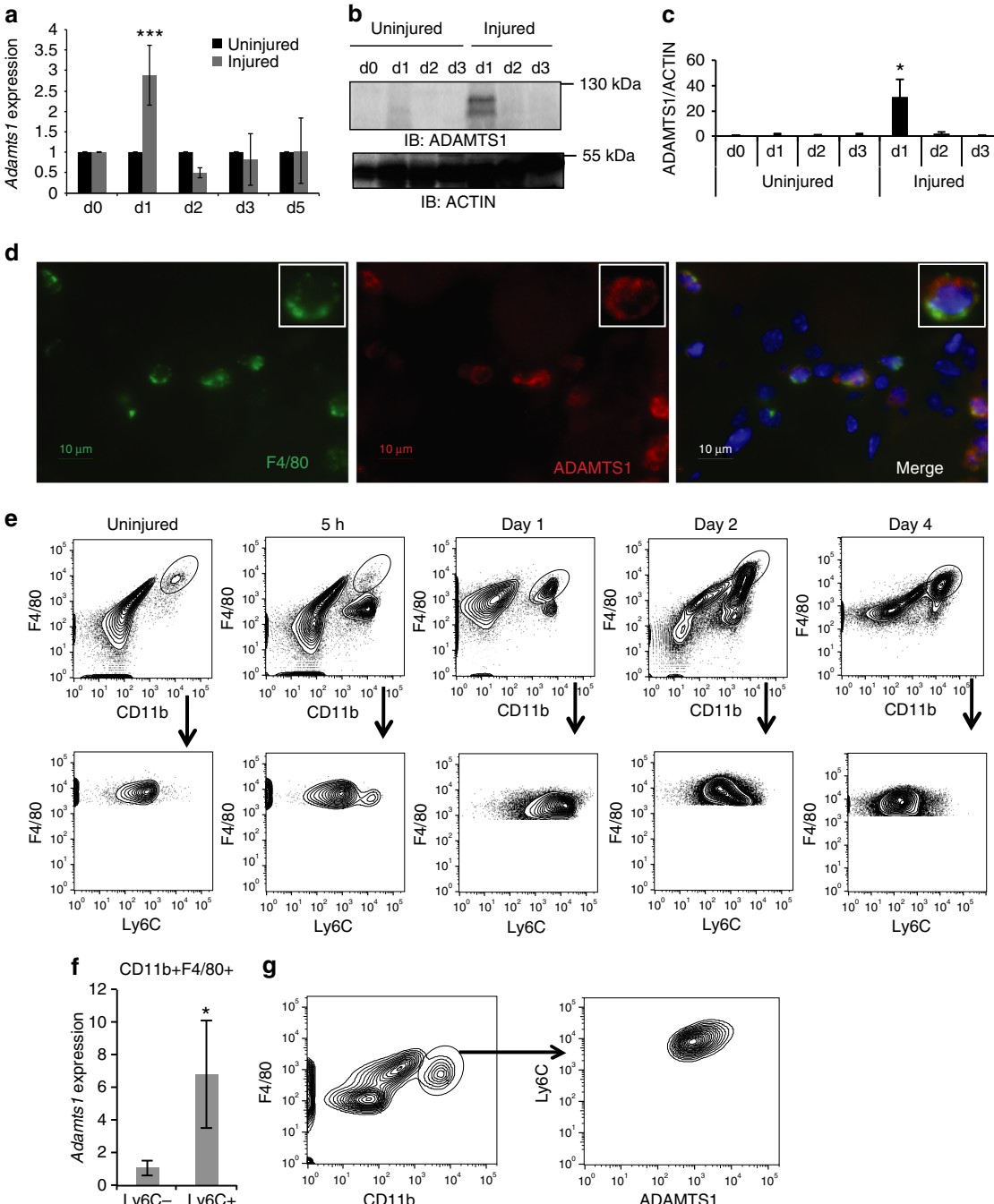

**Fig. 2** Adamts1 is induced by muscle injury and expressed in the macrophages of skeletal muscles. **a** Time-course monitoring the expression level of *Adamts1* in tibialis anterior (TA) muscles isolated from wild-type mice before and after muscle injuries. *Adamts1* levels were quantified using RT-qPCR and the injured muscles were compared to the contralateral uninjured muscle ($n = 5$). **b**, **c** Western blots and quantification of ADAMTS1 protein levels in cell lysates from TA muscles before and after muscle injuries compared to Actin control ($n = 3$ for each time point). **d** Images from IHC performed on frozen sections prepared from TA muscles 1 day post injury (*insets* are higher magnification images of a F4/80- and ADAMTS1-positive cell that is present in the lower magnification field). Histological slides were stained using antibodies against ADAMTS1 (*red*) and F4/80 (*green*); a marker for macrophages. ADAMTS1 co-localized with macrophages (*inset*). *Scale bar* = 10 µm. **e** Flow cytometry over a time-course monitoring Ly6C in F4/80+cells after muscle injuries. **f** RT-qPCR measuring the expression levels of *Adamts1* in Ly6C+ compared to Ly6C− macrophages. **g** Representative flow cytometry plots showing ADAMTS1 expression in the macrophage population isolated from injured muscle tissue. *$P < 0.05$, ***$P < 0.001$. *Error bars* represent s.d. Statistical significance tested using paired *t*-tests

of this ADAMTS1 activity is by targeting NOTCH1 protein on the satellite cells. These findings significantly enrich our understanding of the extracellular signals that regulate satellite cell activation and identify a pathway that could potentially be targeted with therapeutics to enhance muscle regeneration.

## Results

**ADAMTS1 promotes satellite cell activation**. Expression profiling comparing quiescent to activated satellite cells identified a number of genes with previously unknown roles in satellite cell activation[15], implicating a potential role for the product of these

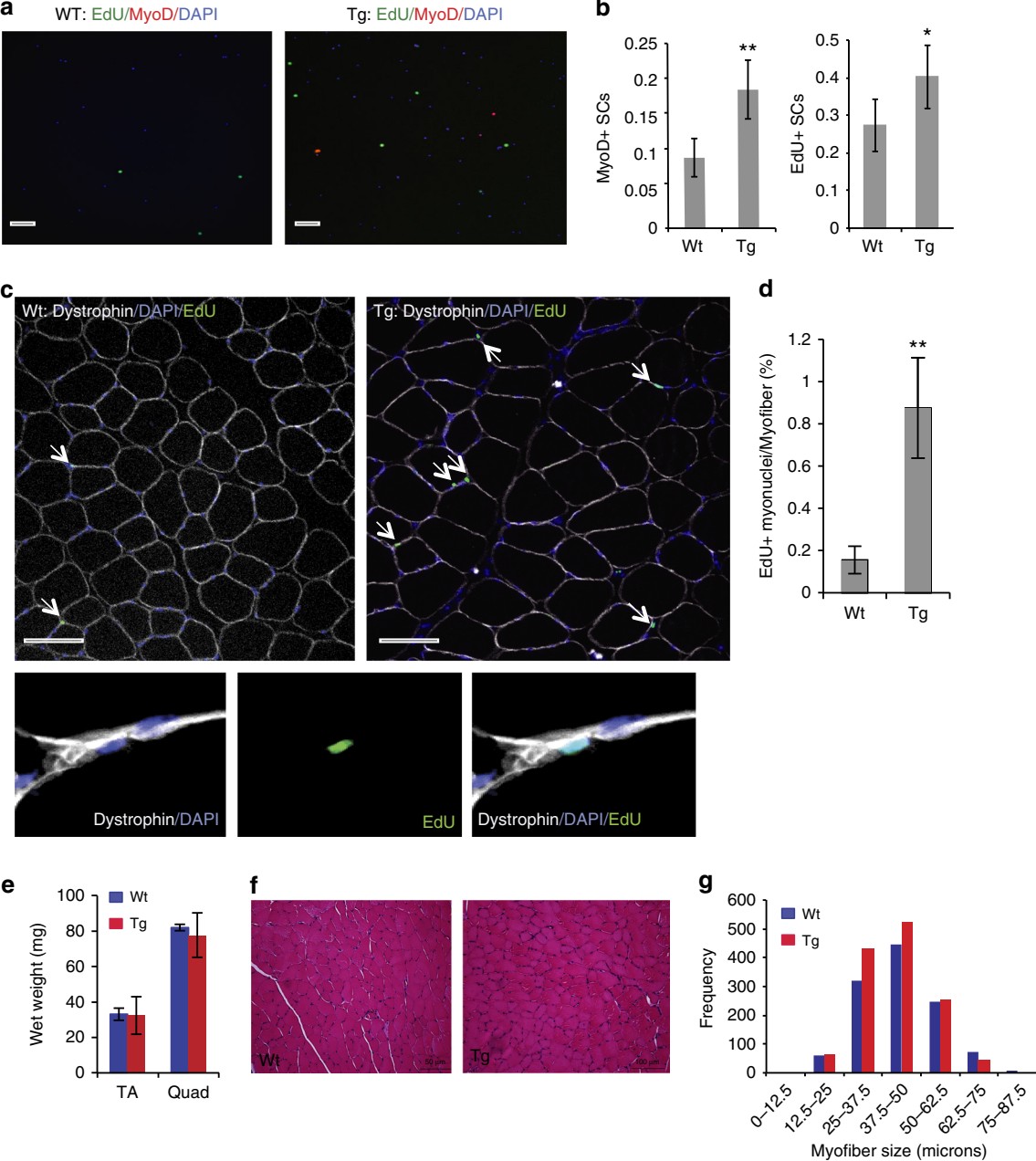

**Fig. 3** Overexpression of Adamts1 promotes satellite cell activation in vivo. **a** Images from immunocytochemistry (ICC) performed on satellite cells immediately following isolation by FACS from wild-type (Wt) and Adamts1 (Tg) mice that had received an in vivo pulse of EdU. Cells were stained with DAPI (*blue*) to identify nuclei and EdU (*green*) and MyoD (*red*) to distinguish activated satellite cells (*scale bar* = 100 μm). **b** Quantification of the percent of satellite cells that are MyoD (*left*) and EdU (*right*) positive in Wt (*n* = 6) and Tg (*n* = 5) mice. **c** (*upper*) Images from IHC performed on histological sections prepared from TA muscles from Wt and Tg animals that had received an in vivo pulse of EdU and stained for DAPI (*blue*), EdU (*green* and *arrows*) and Dystrophin (*white*). *Scale bar* = 100 μm. (*lower*) Higher power magnification images of IHC showing EdU-positive cells (*green*) located beneath the myofiber membrane as delineated by Dystrophin (*white*) staining. **d** Quantification of the number of EdU-positive myonuclei per myofiber in Wt (*n* = 3) and Tg (*n* = 3) mice (>200 myofibers per mouse were examined). **e** Gross wet weights of tibialis anterior (TA) and quadriceps (Quad) muscles in Wt (*n* = 3) and Adamts1 transgenic (Tg; *n* = 4) mice at 1 month of age. **f** Histological sections prepared from TA muscles of 1-month-old mice and stained with hematoxylin and eosin (H&E). *Scale bar* = 50 μm. **g** Quantification of the diameters of myofibersin TA muscles from 1-month-old Tg (*n* = 4) and Wt (*n* = 3) mice. *$P < 0.05$, **$P < 0.01$. *Error bars* represent s.d. Statistical significance tested using paired *t*-tests

genes in the regenerative process. Among these genes, *Adamts1* was particularly intriguing since it lacks the epidermal growth factor-like transmembrane and cytoplasmic modules that tether ADAM proteins to the cell membrane and is secreted[16]. Therefore, we hypothesized that it could participate in coordinating the signal from muscle injury to satellite cell activation. *Adamts1* was previously found to have roles in ovulation,

angiogenesis and cancer[17, 18]. However, a role for *Adamts1* in the regulation of Notch signaling or satellite cell activation was unknown. In order to test if extracellular ADAMTS1 affects satellite cell activation, we treated intact mouse myofibers (where satellite cells remain in their physiological location) with recombinant ADAMTS1 (rADAMTS1) and examined the effect on satellite cells using immunohistochemistry (IHC). These

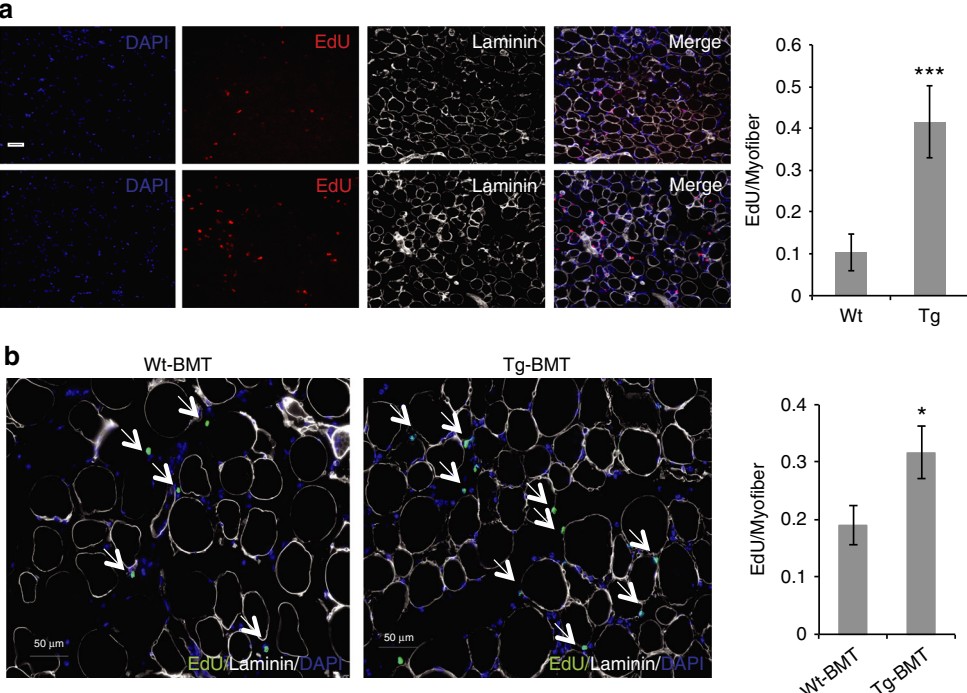

**Fig. 4** Adamts1 mice have increased satellite cell activity in response to injury. **a** (*left*) Images from IHC performed on histological sections prepared from injured TA muscles of 1-month-old Wt and Tg mice. EdU was injected into the mice at the time of muscle injury and mice were sacrificed for IHC on day 1. Histological sections were stained with DAPI (*blue*) to identify cell nuclei and EdU (*green*) to distinguish proliferating satellite cells. *Scale bar* = 100 μm. **a** (*right*) Quantification of the number of proliferating satellite cells after a muscle injury in Wt (*n* = 3) and Tg (*n* = 3) (>200 myofibers per mouse were examined). **b** (*left*) Images from IHC performed on histological sections prepared from injured TA muscle from Wt mice that received bone marrow transplantations from Wt mice (Wt-BMT) or Adamts1 mice (Tg-BMT) donors. *Scale bar* = 50 μm. **b** (*right*) Quantification of the number of proliferating satellite cells after a muscle injury in Wt-BMT (*n* = 3) and Tg-BMT (*n* = 4) mice (>200 myofibers per mouse were examined). *$P < 0.05$, ***$P < 0.001$. *Error bars* represent s.d. Statistical significance tested using paired *t*-tests

studies demonstrate that exposing wild-type myofibers to rADAMTS1 promotes the activation of satellite cells (Fig. 1a–c).

**Muscle injury increases ADAMTS1 levels in wild-type mice.** Satellite cell activation is required for muscle regeneration following injury[1]. As our results indicate that ADAMTS1 promotes satellite cell activation, we examined the pattern of expression of *Adamts1* during muscle regeneration in vivo. First, we monitored *Adamts1* expression in mice over a time course following muscle injury. We found that wild-type mice have a robust induction of *Adamts1* levels in injured muscle 1 day after the injury (Fig. 2a), corresponding to the time when satellite cells begin to break quiescence and enter the cell cycle[19]. We also found that ADAMTS1 protein levels in the injured muscle tissue increase in parallel with the mRNA expression after injury (Fig. 2b, c). However, ADAMTS1 protein is not induced in satellite cells by muscle injury (Supplementary Fig. 1a). To identify the cellular origin of the increased levels of ADAMTS1 in the muscle tissue after injury, we performed IHC on muscle tissues. We discovered that ADAMTS1 protein strongly co-localizes with macrophages infiltrating the site of injury in the muscle tissue (Fig. 2d). Further analysis of the macrophage population in muscles revealed that the Ly6C[+] subtype of macrophages, which are rapidly recruited to sites of muscle injury with a peak at day 1 post injury (Fig. 2e and refs [20, 21]), express high levels of *Adamts1* after muscle injury (Fig. 2f, g and Supplementary Fig. 1b). ADAMTS1 is also expressed by macrophages present in uninjured muscle tissue (Supplementary Fig. 1c) and, intriguingly, macrophages were frequently found located in close proximity to satellite cells in uninjured muscle

(Supplementary Fig. 1d). However, as expected, there are significantly fewer macrophages present in uninjured compared to injured muscle tissue (Supplementary Fig. 1e).

**In vivo effects of ADAMTS1 on satellite cells.** In order to model the ADAMTS1 levels in muscle tissue we observed after injury and specifically test whether increasing the levels of ADAMTS1 can induce satellite cell activation out of quiescence in vivo, we generated transgenic mouse lines that constitutively overexpress *Adamts1* in macrophages (Adamts1 mice). The transgene contains full-length *Adamts1* complementary DNA (cDNA) with a human influenza hemagglutinin-tag sequence subcloned behind aP2 promoter and enhancer elements. Three founder lines were established, which all had similar phenotypes (see Methods for further details on the mouse model). To elucidate the effect of Adamts1 on the satellite cell quiescence in vivo, we conducted 5-ethynyl-2′-deoxyuridine (EdU) pulse-chase experiments[12]. We injected EdU into the mice for 2 weeks and then isolated satellite cells and performed immunocytochemistry (ICC) to quantify the number of EdU- and MyoD-positive cells. We found that Adamts1 mice have significantly more EdU-positive and MyoD-positive satellite cells, indicating that satellite cells in Adamts1 mice are breaking quiescence and spontaneously activating at a higher frequency than in wild-type littermate animals (Fig. 3a, b).

Next, we traced the fate of the activated satellite cells in vivo. As EdU labels only dividing cells, EdU-positive myonuclei arise only from the fusion of activated proliferating satellite cells that differentiated during the pulse-chase period. We found that the number of EdU-positive myonuclei is dramatically increased in

muscles of Adamts1 mice compared to wild-type littermates (Fig. 3c, d).

When we examined the muscles of 1-month-old Adamts1 mice, we did not detect any significant differences in either muscle wet weight or myofiber size compared to sex- and age-matched wild-type littermate controls (Fig. 3e–g), indicating a negligible effect on embryonic and postnatal muscle

progenitors. At 4 months of age, Adamts1 mice developed increased muscle mass with larger myofibers, increased numbers of myonuclei per myofiber and increased total body weights (Supplementary Fig. 2a–f), a finding consistent with an effect of Adamts1 on quiescent satellite cells.

Within hours of a muscle injury, neutrophils and macrophages infiltrate into the wounded site and are thought to play a role in

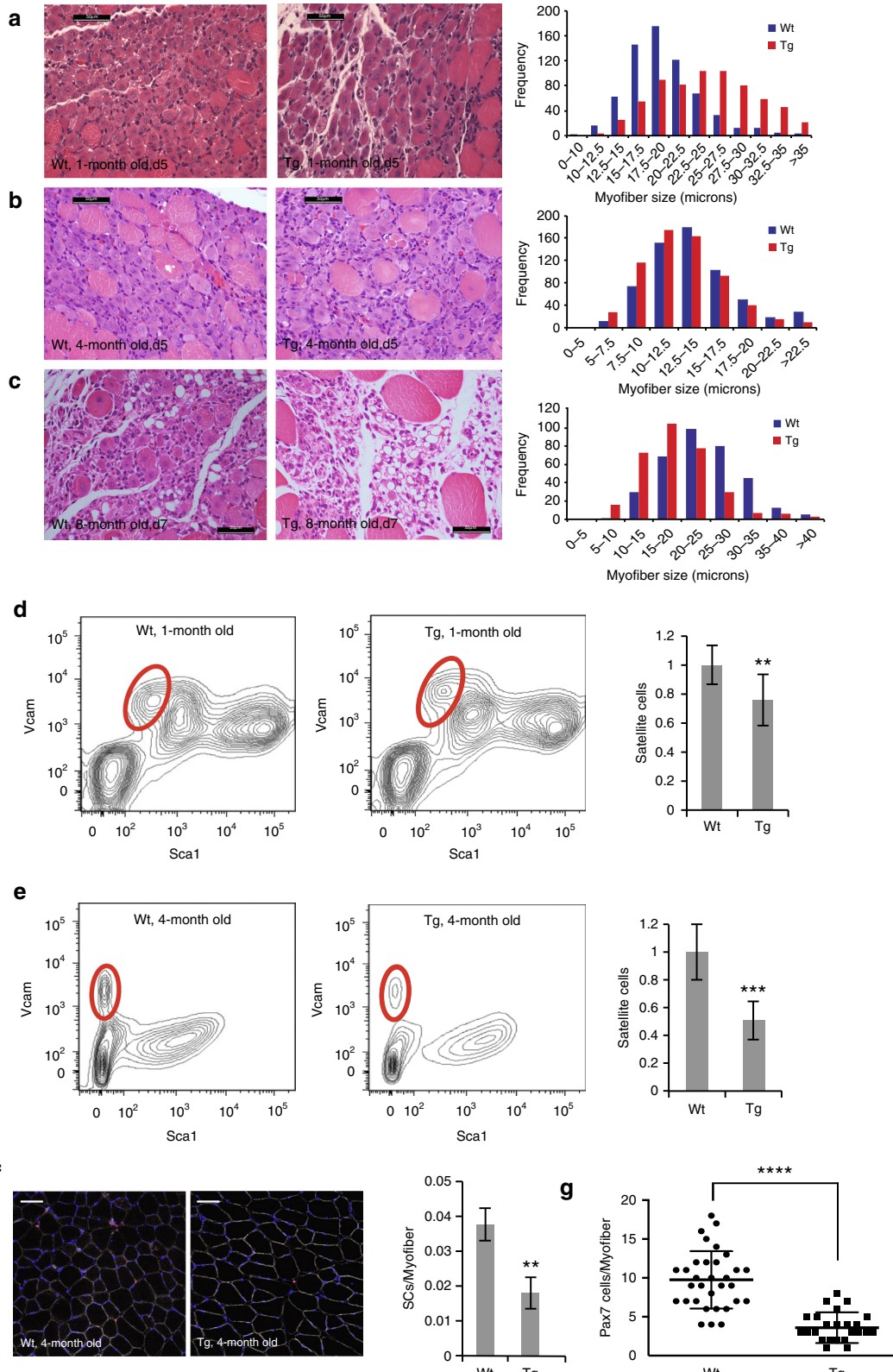

clearing the cellular debris and contributing to muscle regeneration[22]. Given the increased expression of *Adamts1* in infiltrating macrophages and the effects ADAMTS1 on satellite cells, we hypothesized that Adamts1 mice would have increased levels of satellite cell activation after a muscle injury compared to wild-type littermates. To test this hypothesis, Adamts1 and wild-type littermate mice were injected with EdU in vivo at the same time as a tibialis anterior (TA) muscle injury and IHC was performed on the injured muscle on day 1 after injury to quantify the number of EdU-positive satellite cells. We found that Adamts1 mice have increased levels of satellite cell proliferation after muscle injury (Fig. 4a), consistent with our findings of enhanced satellite cell activation.

**ADAMTS1 is secreted from macrophages in injured muscle**. To rigorously test if the satellite cell phenotype in Adamts1 mice is primarily caused by macrophage-secreted ADAMTS1, we performed a series of experiments. First, we quantified the expression level of *Adamts1* in macrophages, satellite cells and fibro–adipogenic precursor cells from the muscles of injured and uninjured Adamts1 mice compared to wild-type mice at 1 and 4 months of age. These studies confirmed that macrophages are the primary cell type overexpressing *Adamts1* in the Adamts1 mice (Supplementary Fig. 3a). Next, we conducted tissue culture experiments using primary satellite cells isolated from wild-type mice cultured alone or co-cultured with macrophages from wild-type mice or Adamts1 mice followed by ICC for MyoD to monitor the level of satellite cell activation. These experiments revealed that co-culturing wild-type satellite cells with macrophages increases satellite cell activation and macrophages from Adamts1 mice significantly enhance this process (Supplementary Fig. 3b). Finally, we tested the role of macrophages in causing the satellite cell phenotype in vivo. We performed bone marrow transplantation (BMT) studies using donor bone marrow from Adamts1 mice or wild-type control mice transplanted into wild-type recipient mice that had undergone radioablation of their bone marrow. After engraftment of the donor bone marrow, muscles of recipient mice were injured and the mice were pulsed with EdU. While the magnitude of the phenotype was somewhat reduced post BMT (possibly because of secondary effects of the radioablation), wild-type mice that received bone marrow from the Adamts1 mice have significantly higher levels of EdU-positive satellite cells post injury than wild-type mice that received wild-type bone marrow (Fig. 4b). Together, these findings indicate that macrophage-secreted ADAMTS1 is responsible for the satellite cell phenotype.

**Implications for persistent satellite cell activation**. We next examined the impact of the enhanced level of satellite cell activity on muscle regeneration after injury. Intriguingly, we discovered

that 1-month-old Adamts1 mice have an accelerated recovery from muscle injury compared to wild-type littermate controls (Fig. 5a). However, this phenotype is lost by the time the Adamts1 mice are 4 months of age (Fig. 5b). By the time the mice reach 8 months of age, regeneration in Adamts1 mice is mildly impaired (Fig. 5c). We speculated that the progression of this phenotype was caused by persistent satellite cell activation in the Adamts1 mice, depleting the satellite cell pool. To test this hypothesis, we quantified the satellite cell population as the mice aged. Flow cytometry and IHC confirmed that at 1 month of age, the satellite cell population in Adamts1 mice is reduced to ~75% of control values and by 4 months of age, it is reduced to ~50% of control values (Fig. 5d–g). These results implicate the progressive exhaustion of the satellite cell pool from persistent activation as the cause of impaired regeneration with aging in the Adamts1 mice despite the enhancement of satellite cell activation by Adamts1.

**ADAMTS1 inhibits Notch signaling**. The satellite cell phenotype of the Adamts1 mice resembles that of mice in which Notch signaling is deleted in the satellite cell population[5, 6]. Furthermore, the increase of Adamts1 levels in injured muscle (Fig. 2a–c) corresponds to the timing of satellite cell activation, a process that requires a decrease in Notch signaling[5, 6]. Indeed, we confirmed this reduction in Notch signaling by demonstrating a decrease in the expression levels of Notch1 target genes *Hes-1* and *Hey-1* (Fig. 6a, b).The later increases in expression of these Notch targets are associated with the second phase of Notch signaling during which Notch promotes the proliferative expansion and inhibits the differentiation of transit-amplifying cells[4].

To further explore the relationship between Adamts1 and Notch1 signaling in quiescent satellite cells, we compared the levels of the active NOTCH1 intracellular domain (NICD) in satellite cells isolated from Adamts1 mice and wild-type littermates. We discovered that satellite cells directly isolated from Adamts1 mice have lower levels of NICD than wild-type littermates (Fig. 6c, d). We also found that satellite cells isolated from Adamts1 mice have lower levels of expression of Notch target genes (Fig. 6e), indicating reduced Notch signaling. Together, these data suggest that Adamts1 might induce satellite cell activation by suppressing Notch1 signaling.

NOTCH1 is a transmembrane protein, thus providing a site for interaction with extracellular signals[23]. Using co-immunoprecipitation (co-IP), we found that ADAMTS1 interacts with the transmembrane/intracellular component (TMIC) of NOTCH1 in muscle tissues of Adamts1 mice (Fig. 6f). Next, we generated a series of Adamts1 deletion mutant expression constructs (Fig. 6g) and mapped the NOTCH1 interaction domain to the metalloproteinase domain of ADAMTS1 (Fig. 6h). In addition, we

**Fig. 5** Adamts1 mice have accelerated regeneration when young but a depleted satellite cell pool with aging. **a–c** (*left*) Images of H&E-stained histological sections prepared from TA muscle injuries that were induced at 1 **a**, 4 **b** and 8 **c** months of age in Wt and Tg mice (mice were sacrificed 5 days post injury for the 1- and 4 month-old mice and 7 days post injury for the 8-month-old mice). *Scale bar* = 50 μm. **a–c** (*right*) Quantification of the cross-sectional myofiber diameters post injury calculated for **a** 1-month-old (Wt: $n = 6$; Tg: $n = 3$), **b** 4-month-old (Wt: $n = 4$; Tg: $n = 4$) and **c** 8-month-old (Wt: $n = 3$; Tg: $n = 3$) mice. The numbers of myofibers counted is displayed in the frequency axis. The 1-month-old Tg mice had accelerated regeneration compared to Wt littermates ($P < 0.0001$). The 8-month-old Tg mice had impaired regeneration compared to Wt littermates ($P < 0.0001$). **d, e** (*left*) Flow cytometry plots of CD45−/CD31−/Scal1−/Vcam+ satellite cells in Tg and Wt mice at **d** 1 and **e** 4 months of age. **d, e** (*right*) Quantification of the number of satellite cells per total live mononuclear cells isolated (~ $5 \times 10^5$ cells from each animal were analyzed) from the hind limb muscles of 1-month-old (Wt: $n = 11$; Tg: $n = 10$) and 4-month-old (Wt: $n = 11$; Tg: $n = 9$) mice showing a depletion of satellite cells in Tg mice (**$P < 0.01$ at 1 month; ***$P < 0.001$ at 4 months). **f** (*left*) Images from IHC of histological sections of TA muscles isolated from 4-month old Wt and Tg mice and stained for Pax7 (*red*), Dystrophin (*white*) and DAPI (*blue*). *Scale bar* = 100 μm. **f** (*right*) Quantification of the number of Pax7-positive satellite cells per myofiber show Tg mice have fewer satellite cells than Wt mice ($n = 3$ for each genotype; >200 myofibers per mouse were examined). **$P < 0.01$. **g**, Quantification of the number of Pax7-positive cells on freshly isolated myofibers from 4-month-old mice (each point represents a myofiber, $n = 3$ mice for each genotype) demonstrated that Tg mice have fewer satellite cells. ***$P < 0.001$. *Error bars* represent s.d. Statistical significance tested using $\chi^2$ for myofiber diameters and paired *t*-tests for other comparisons

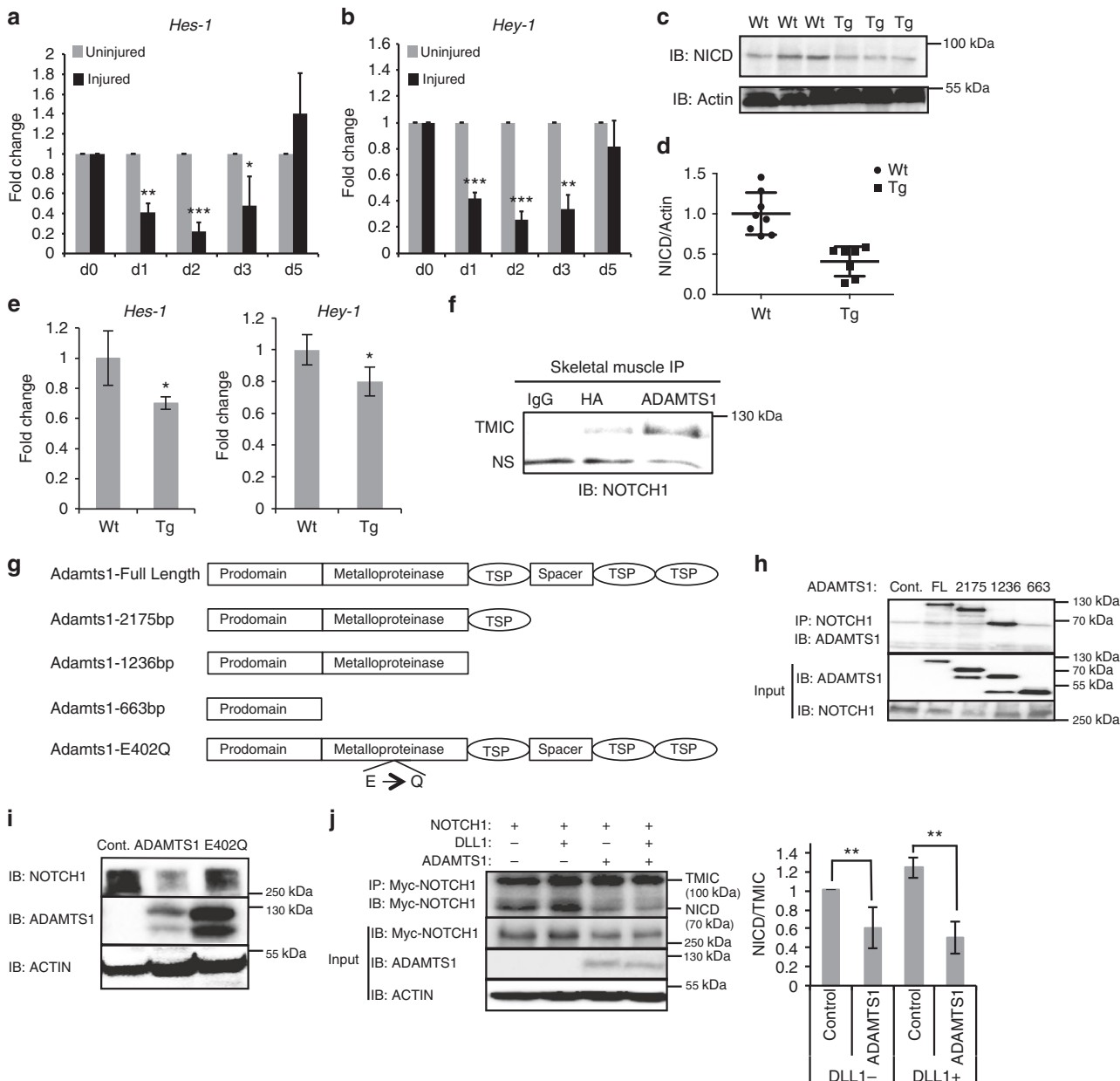

**Fig. 6** Adamts1 inhibits Notch signaling. **a**, **b** Time-course monitoring the expression levels of Notch target genes *Hes1* and *Hey1* in TA muscles isolated from wild-type mice before and after muscle injuries. *Hes1* and *Hey1* levels were quantified using RT-qPCR and the injured muscles were compared to the contralateral uninjured muscles ($n = 5$ mice per time point). **c** Representative western blot of lysates from primary myoblasts isolated from Tg and Wt mice. NICD=Notch1 intracellular domain (activated form of NOTCH1). Actin was used as a loading control. **d** Quantification of the levels of NICD protein in Tg ($n = 7$) compared to Wt ($n = 8$). **e** Quantification of *Hes1* and *Hey1* expression levels using RT-qPCR on primary satellite cells isolated from Wt ($n = 3$) and Tg ($n = 3$) mice by FACS. **f** Co-immunoprecipitation (IP) of ADAMTS1 and NOTCH1 from primary muscle tissue isolated from Tg mice using anti-HA (transgene has a hemagglutinin (HA) tag), anti-ADAMTS1 or IgG control antibodies. The transmembrane/intracellular fragment of NOTCH1 (TMIC) was immunoprecipitated specifically with ADAMTS1. NS is a nonspecific band. **g** Schematic of deletion and mutant Adamts1 expression constructs generated. **h** NOTCH1 co-IPs with ADAMTS1 and sequential ADAMTS1 deletion mutants map the NOTCH1 interaction domain to the metalloproteinase domain. Cont, control empty plasmid; FL, full-length Adamts1. **i** Western blots on cell lysates from co-cultured cells transfected with *Notch1* and either *Adamts1* (ADAMTS1) or E402Q *Adamts1* (E402) or empty plasmid control (Control) expression plasmids. **j** (*left*) IP of processed myc-tagged NOTCH fragments from co-cultures with cells overexpressing *Adamts1*, *Notch1* and *Dll1* or empty vector controls. **j** (*right*) Quantification of the levels of NICD/TMIC fragments in response to co-culturing with *Adamts1* overexpressing cells with and without cells expressing *Dll1*. $P < 0.05$, **$P < 0.01$, ***$P < 0.001$. *Error bars* represent s.d. Statistical significance tested using paired *t*-tests

discovered that co-culturing cells that have been transfected with *Adamts1* and *Notch1* expression plasmids leads to a dramatic decrease in NOTCH1 levels, a decrease that is dependent on the metalloproteinase activity of Adamts1 (Fig. 6i), indicating that NOTCH1 is a target of ADAMTS1 proteinase activity.

We also examined the effect of ADAMTS1 on NOTCH1 processing. Cleavage of NOTCH1 at site 2 (S2), which is located in the extracellular TMIC domain, results in processing of TMIC to an active NICD fragment[24]. Since the level of NICD is reduced in satellite cells from Adamts1 mice (Fig. 6c, d), we tested the

effect of ADAMTS1 on processing of NOTCH1 at the S2 cleavage site. We co-cultured cells overexpressing Myc-tagged NOTCH1 with and without overexpression of ADAMTS1 and with and without cells overexpressing the NOTCH1 ligand Delta1 (DLL1). We then immunoprecipitated the Myc-tagged processed NOTCH1 fragments from the co-cultured cells. These studies revealed that ADAMTS1 inhibits normal processing of NOTCH1 at the S2 cleavage site and that this activity is not rescued by overexpressing DLL1 (Fig. 6j).

## Discussion

These studies identify ADAMTS1 as an extracellular protein that induces satellite cell activation. Our finding that macrophages are a source of ADAMTS1 during regeneration from muscle injury in wild-type mice is particularly intriguing. In spite of potential deleterious effects on muscle tissue from chronic disease[25], macrophages are critical for muscle regeneration after injury[20, 26]. Furthermore, prior work by others[20, 27, 28] demonstrated that the acute role of macrophages immediately following muscle injury has sustained implications for the regenerative process. However, the mechanisms by which macrophages promote proper muscle regeneration have remained incompletely understood. Our results indicate that secretion of ADAMTS1 from macrophages following a muscle injury stimulates satellite cells to break quiescence. Young Adamts1 mice have accelerated regeneration following muscle injury, raising optimism that this pathway could potentially be targeted for a variety of therapeutic opportunities related to muscle injury or dysfunction. However, it is of important cautionary note that we also found that the satellite cell pool is depleted with aging in Adamts1 mice, which was associated with impaired muscle regeneration.

We also show that ADAMTS1 inhibits Notch1 signaling in satellite cells. NOTCH1 is an established regulator of satellite cell quiescence;[5, 6] however, the factors that inhibit Notch signaling and promote satellite cells to break quiescence remain poorly defined. Furthermore, the impact of Notch signaling on muscle tissue in general is complex, including important stage-specific and context-dependent parameters[29] that influence satellite cells, muscle mass and regenerative capacity. These caveats raise the possibility that multiple signals or effects on context-specific factors are also contributing to the phenotype in the Adamts1 mice and studies using a combination of targeted genetic deletions of Notch1 and Adamts1 are needed to identify and distinguish other context-specific roles and the contribution of each context to overall muscle physiology. However, the data described here indicate that macrophage-secreted ADAMTS1 has an important role in the regulation of satellite cell activation.

It is interesting that cell membrane-bound metalloproteinases ADAM10 and ADAM17/TACE are critical for processing Notch[13, 30], resulting in activation of Notch signaling in a variety of contexts, including satellite cells. Our results provide compelling evidence that ADAMTS1, a secreted metalloproteinase not previously known to have a role in satellite cell activation, also targets NOTCH1 but, importantly, has the opposite effect and inhibits Notch signaling. We are not aware of any prior reports identifying specific proteases that inhibit, rather than facilitate, Notch signaling. Therefore, our results reveal a novel mechanism for targeting and inhibiting Notch signaling that promotes satellite cell activation by a macrophage-secreted metalloproteinase. It will be of broad interest to elucidate additional details of the specific molecular events that coordinate this inhibition of Notch signaling.

In summary, our results reveal that Adamts1 is a previously unrecognized potent modulator of NOTCH1 activity that is triggered during muscle regeneration to stimulate satellite cell activation. Together, these data enhance our understanding of the regulation of satellite cell activation and regeneration after injury.

## Methods

**Animals.** Wild-type mice were purchased from Jackson Laboratory at the ages indicated to perform the studies on isolated satellite cells, intact myofibers and muscle injuries. Pax7[CreER/CreER] mice were provided by Charles Keller (Oregon Health and Science University) and ROSA26[eYFP/eYFP] mice were purchased from Jackson Laboratory. For the Adamts1 mice, the transgene was generated by subcloning the full-length Adamts1 cDNA under an aP2 promoter/ enhancer element followed by the hGH polyadenylation sequence. Three founder lines were generated that all had a similar phenotype and one founder line was selected for more detail analysis. The transgenic mice were fertile, with no gross developmental abnormalities, and transgenic progeny were generated in the expected Mendelian ratios. In addition to macrophages, aP2 is expressed prenatally in ganglia, cartilage and vertebrae and postnatally in adipose tissue;[31] we did not detect any morphological, histological or functional changes in the ganglia, cartilage or vertebrae of the transgenic mice. There was an ~25% decrease in adipose tissue mass in the Adamts1 mice.

Sex-matched littermates were used as wild-type controls for the experiments using Adamts1 mice. All experiments were approved by the institutional animal care and use committees and performed in accordance with Stanford University institutional guidelines.

**Bone marrow transplantation.** Bone marrow was harvested from 3-month-old Adamts1 and wild-type littermate donor mice and $3\times10^6$ cells were transplanted by tail-vein injection into lethally irradiated (9 Gy) wild-type recipient mice. Engraftment of donor bone marrow from Adamts1 mice was confirmed by quantitative PCR of DNA isolated from the peripheral blood of the wild-type recipient mice 2 months after transplantation using transgene-specific primers (forward primer: 5′-TTAATGGACACCCTGCTTCC-3′; reverse primer: 5′-ATCTTCCCATTCTAAACAACACCC-3′). Muscle injuries were performed on recipient mice as described below.

**Satellite cell isolation and culture.** Hind limb muscles were dissected, finely minced and dissociated by digestion with Collagenase II (0.2%) in Ham's F10 medium containing 10% house serum for 90 min. Then the fragmented myofibers were washed and further digested in Collagenase II (0.2%) and dispase (2.5 units per ml; Invitrogen) for an additional 30 min. Digested fiber suspensions were triturated to yield a mononuclear cell suspension. Mononuclear cells were stained with Vcam-biotin (BD Bioscience, 1:50, clone 429), Streptavidin-APC was used to amplify the Vcam Signal (BD Bioscience, 1:75, clone 17-4317-82), Sca1-FITC (BD Bioscience, 1:100, clone D7), CD31-PE-cy7 (BD Bioscience, 1:100, clone 390) and CD45-PE-Cy7 (BD Bioscience, 1:100, clone 30-F11). Fluorescence-activated cell sorting (FACS) was performed on the stained cells using a BD FACSAria II cell sorter to isolate CD31−/CD45−/Sca1−/Vcam+ satellite cells using a previously described gating strategy[32] and detailed in Supplementary Fig. 4. For satellite cell culture experiments, sorted satellite cells were plated on Matrigel (Sigma, 1:100)-coated dishes and cultured in Dulbecco's modified Eagle's medium (DMEM) with 20% fetal bovine serum (FBS), 10% horse serum (HS) and 3% chicken embryo extract (CEE). To differentiate satellite cells, cells were cultured in DMEM with 2% HS.

**Histology and immunohistochemistry.** TA muscles were dissected for hematoxylin and eosin staining. Myofiber size was determined by measuring the diameter of TA muscle fibers using ImageJ. For immunohistochemistry, TA muscles were fixed for 5 h using 0.5% paraformaldehyde and subsequently transferred to 20% sucrose overnight. Muscle was then frozen in Optimum Cutting Temperature (OCT) Compound, cryosectioned with a thickness of 6 μm and stained using M.O.M kit (Vectorlabs) and EdU visualization kit (Click-it, Invitrogen). Images were taken of cross-sections of Dystrophin and EdU immunostained TA muscles from 3 mice at each age. The antibodies used in this immunostainig were: MyoD (Dako and BD Biosciences, 1:1000, clone MS-273-P), Dystrophin (1:1000, Sigma, clone MANDRA1) and Pax7 (1:50, DSHB, clone 4ea). To generate the Pax7 antibody, supernatant media from cultured Pax7 hybridoma cells were collected and concentrated through centrifugal concentration column (0.5 ml, Millipore).

**Macrophage analysis, isolation and IHC.** Muscle tissue was isolated from the mice and the fascia was sacrificed. Mononuclear cells were isolated from hind limb muscles that were finely minced and enzymatically digested with 0.2% collagenase II as described above in the satellite cell isolation section. The mononuclear cells were stained with the following antibodies: Ly6G-V450 (BD Bioscience, 1:75, clone1A8), F4/80-FITC (Biolegend, 1:75, clone BM8), CD11b-APC-Cy7 (BD Bioscience, 1:100, clone M1170) and Ly-6C-APC (Ebioscience; 1:75, clone 4K1.4). To quantify the level of ADAMTS1 in macrophages by flow cytometry, cells were co-stained with ADAMTS1 antibody (R&D Systems, 1:50, clone CDSL0115031). To quantify ADAMTS1 levels in macrophages after muscle injury, injured muscles

were harvested day 1 post injury. Cells were analyzed and purified by FACS using a BD FACSAria II.

To confirm the co-localization of ADAMTS1 with macrophages by histology, cryosections of TA muscles were stained with F4/80-FITC (Biolegend, 1:75, clone BM8) and ADAMTS1 antibody (1:500 Santa Cruz Biotech, 1:500, clone A19) at 4 °C overnight. The sections were washed and incubated with fluorescent-dye conjugated secondary antibodies (1:1000, Invitrogen) for 30 min at room temperature. The sections were covered with antifade mounting medium (Vector Laboratories) and imaged.

**Cloning and mutagenesis.** Full-length *Adamts1* cDNA was generated by reverse transcription-PCR (RT-PCR) of RNA isolated from mouse adipocytes and the amplicon was subcloned into a mammalian expression vector (pEF/v5-HisTOPO, Invitrogen) and confirmed by sequencing. The various Adamts1 deletions were generated as detailed in the manuscript. Site-directed mutagenesis was performed to generate the E402Q metalloproteinase-dead mutant. Full-length mouse Notch1 with 6X Myc-tag was obtained from Addgene (Plasmid 41728: pCS2 Notch1 Full Length-6MT).

**ADAMTS1 recombinant protein production and purification.** rADAMTS1 was purchased (R&D Systems) or purified from the supernatant of CHO cells over-expressing the protein. To generate CHO cells overexpressing ADAMTS1, CHO cells from American Type Culture Collection (ATCC) were transfected separately with either a control plasmid or the full-length His- and V5-tagged *Adamts1* expression plasmid by Lipofectamine 2000 (Invitrogen). Single-cell isolates were used to establish clonal stable CHO cell lines using selection with 5 µg/ml blasticidin. Overexpression of ADAMTS1 was verified by western blot. The stable clonal CHO cell lines were cultured in DMEM containing 10% FBS until confluence. Then, the culture medium was changed to Opti-MEM (Thermo Fisher) 2 days prior to collection of the cell culture supernatants. ADAMTS1 protein was purified from the culture by running the supernatant through aNi-NTA agarose column (MCLAB) where protein was allowed to bind for 1 h at room temperature with gentle agitation followed by washing and elution. The purified proteins were finalized by dialysis at 4 °C overnight in phosphate-buffered saline (PBS) containing 20% glycerol.

**Muscle fiber isolation and culture.** To isolate single myofibers, extensor digitorum longus muscles from tamoxifen-treated (5 mg/injection, one injection every third day for a total of 5 treatments) Pax7$^{CreER/+}$;Rosa26$^{eYFP/+}$ animals were dissected and subjected to enzymatic digestion with collagenase I (2 mg/ml, Invitrogen) for 90 min at 37 °C in Ham's F10 medium containing 10% HS. The single myofibers from muscles were gently released by triturating muscle tissues against the wall of 100 mm plates. Then, undamaged fibers were transferred into new dishes to remove debris and interstitial cells[15]. For culture experiments, myofibers were maintained in proliferative medium (20% FBS, 10% HS, 3% CEE) containing rADAMTS1 or control proteins. Myofibers were fixed in 4% paraformaldehyde in PBS at 6 and 72 h following the addition of rADAMTS1 or control proteins. Standard myofiber immunostaining was performed[15] using antibodies for MyoD (BD Biosciences, 1:100, clone MS-273-P) and green fluorescent protein (Abcam, 1:100, 1:100, ab 13970) to detect the yellow fluorescent protein (YFP) reporter signal.

**Muscle injury.** Mice were anesthetized using isoflurane through a nose cone. Muscle injuries were induced by injecting 50 µl of 1.2% BaCl$_2$ into the TA muscle[5]. In EdU tracing experiments, mice were given intraperitoneal injection of 200 µl EdU (4 mg/ml) at the time of injury.

**Co-immunoprecipitation and western blot analysis.** HEK-293T (ATCC) cells were co-transfected with *Adamts1* and *Notch1* expression plasmids. The co-immunoprecipitation was performed using magnetic dynabeads protein G (Invitrogen). In co-culture experiments, HEK-293T cells expressing NOTCH1 and ADAMTS1 were co-cultured with NIH-3T3 cells (ATCC) expressing DLL1 for 5 h. The cell lysate was analyzed by co-immunoprecipitation and western blot. For western blotting, the muscle tissues and cells were extracted in RIPA lysis buffer (1× PBS, 1% NP-40, 0.1% SDS, 0.5% sodium deoxycholate, 1 mM EDTA). Protein extracts were subjected to electrophoresis on polyacrylamide gels and transferred to nitrocellulose membranes. The membranes were first incubated in blocking buffer and then incubated with antibodies. The antibodies used in this study were against: β-actin (Sigma, 1:5000, clone C2), NOTCH1 (BD Pharmingen, 1:200, clone MN1A), cleaved NOTCH1 val 1744 (Cell Signaling, 1:1000, clone 2421S), V5 (1:1000, Invitrogen) and ADAMTS1 (R&D Systems, 1:1000, clone CDSL0115031). Uncropped immunoblots are located in Supplementary Fig. 5.

**Quantitative RT-PCR.** Total RNA was isolated from cell or tissue using RNeasy micro kit(Qiagen) or Trizol (Invitrogen). Quantitative RT-PCR was performed using DyNAmo ColorFlash SYBR Green qPCR kit (Thermo Scientific). Primer sequences are listed in Supplementary Table 1.

**Statistical analysis.** Statistical analysis was performed using Gadpad or Excel software, and specific tests are noted in the text and figure legends.

**Data availability.** The authors declare that all the data supporting the findings of this study are available within the article (and Supplementary Files), or available from the corresponding author on reasonable request.

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

## Acknowledgements

We thank Julien Sage (Stanford University) and the members of the Feldman Laboratory for helpful comments and discussions. H.D. was supported by a NIH T32 fellowship (T32 DK007217), the Child Health Research Institute and the Stanford CTSA (UL1 TR000093) and a Glenn Foundation Fellowship. M.N.W. was supported by a NIH K99 Pathway to Independence grant (K99AG053438). A.A.M. was supported by grants from the NIH (AG043235) and the California Institute of Regenerative Medicine (TG2-01159). A.A. was supported by the Child Health Research Institute at Stanford. The research was supported by an NIH Director's New Innovator Award (DP2 OD006740) and R01 DK114217 to B.J.F., The Glenn Foundation for Medical Research and grants from the Department of Veterans Affairs and the NIH (P01 AG036695, R01 AG23806, and R01 AR062185) to T.A.R. B.J.F. is the Bechtel Endowed Faculty Scholar.

## Author contributions

B.J.F. conceptualized the project. B.J.F. and T.A.R. designed the experiments. H.D., C.-H. S., M.N.W., A.A.M., J.C., A.A. and B.J.F. performed the experiments. H.D. and B.J.F. wrote the manuscript with input from all authors.

## Additional information

**Competing interests:** Stanford University and B.J.F. filed patent PCT/US2013/070389 related to the discoveries described in this manuscript. The remaining authors declare no competing financial interests.

