## [Peer Review file · Nature Communications]

Reviewers' comments:

Reviewer #1 (Remarks to the Author):

In the present study, Du et al. showed that 1) ADAMTS1 was transiently expressed after muscle injury by infiltrating macrophages, 2) mutant mice which express ADAMTS under the control of an α P2 promoter showed enhanced muscle regeneration when they are relatively young. However, as they grow old, those mice exhibited delayed muscle regeneration due to the decrease in the satellite cell pool, and 3) ADAMTS1 interacted with NOTCH1 and suppressed Notch signaling through proteolytically processing NOTCH1. The concept of the present study, that the activity of Notch signaling is negatively regulated by proteolytic cleavage by ADAMTS1, is novel and interesting. The study will potentially be of interest to the researchers in the fields of Notch biology and muscle regeneration. On the other hand, the data presented in the present study is not stringent enough to support their conclusions. Therefore, the manuscript cannot be considered suitable for publication in the present form.

Major points

1. Data on the phenotypes of Adamts1-mice is insufficient. Do they grow normally without any apparent defects? It is certain that α P2/FABP4 is highly expressed in macrophages; however, it is also expressed in the adipose tissue, ganglia, cartilage, and vertebrae (at least during embryonic development) (Transgenic Res (2006) 15:647–653). Because ADAMTS1 is a secreted enzyme, overexpression of ADAMTS1 can theoretically have systemic effects. Moreover, as evidenced by various Notch-related gene mutant mice described in past studies, any deterioration in Notch signaling could have a significant impact on cell fate decision, cell growth, etc, in vivo. Given that ADAMTS1 suppresses Notch signaling as the authors claim, it is highly likely that Adamts1-mice exhibit defects that are related to loss of Notch signaling during embryonic development or postnatal growth. Therefore, further characterization of Adamts1 mice is required to validate their conclusions.

2. There are notable differences in the phenotypes between Adamts1 mice described in the present study and other types of mutant mice in which Notch-signaling is conditionally abrogated. For example; mutant mice with defective Notch signaling in satellite cells do not usually develop muscle hypertrophy or show increased body weight, at least under unchallenged conditions; depletion of satellite cells does not result in thinner muscle fibers but in an incomplete muscle regeneration which is highlighted by muscle fatty infiltration and accumulation of fibrous tissues. These discrepancies on the phenotype of Adamts1 mice with that of other Notch signaling mutant mice have to be thoroughly examined and discussed.

3. Figure 5i shows that coexpression of Adamts1, but not proteolytically inactive mutant, with Notch1 suppresses the expression of NOTCH1 protein in vitro. Based on this data (and other data showing a decrease in the transcripts for Hey1 and Hes1 in the Adamts1 mice-derived satellite cells), the authors concluded that ADAMTS1 suppresses Notch-signaling through proteolytically processing NOTCH1 and thereby decreasing its expression. This is an interesting hypothesis, which indicates a previously unknown mechanism in regulating the Notch availability on cell surface. However, the data in the present study (Figure 5i) is insufficient to convincingly support the conclusion. Given the impact of this novel hypothesis, highly stringent and extensive data on the processing of NOTCH1 by ADAMTS1 and the consequence of NOTCH1 cleavage must be presented.

Minor points

1. Figure 1: The number of Pax7- MyoD+ cells may also be shown. Please also provide the concentration of ADAMTS1 used in this experiment.

2. Figure 2a: Why does the expression of ADAMTS1 fully subside on day 2, even though there must be a number of macrophages in the injury site? Do these macrophages cease the production of ADAMTS 48 h after the injury? Please clarify this issue or add a discussion on this.
3. Figure 2d: F4/80 is a membrane bound protein and is mainly expressed on cell surface. However, Figure 2d shows a positive staining of F4/80 exclusively in the nucleus. This requires an explanation. Furthermore, the result does not necessarily show that satellite cells are negative for ADAMTS1, unless co-staining with a satellite cell marker is presented.
4. As mentioned above, since aP2 is not exclusively expressed in macrophages, bone marrow transfer experiments should be performed to confirm that ADAMTS1 produced by macrophages are responsible for the phenotypes of Adamts1 mice described in the present study.
5. Figure 4g aims to show a decrease in the population of VCAM+ SCA1- cells in Tg mice compared to Wt mice. However, a significantly more number of cells was analyzed in Wt mice compared to Tg mice. An equal number of cells between Tg and Wt has to be analyzed to present proportional increase or decrease in the number of satellite cells.
6. Figure 4c-e: The peak of the histograms among 1-, 4-, and 8-month old WT mice differs significantly (15-17.5, 12.5-15, and 20-25, respectively). This requires an explanation.
7. Flow cytometry data in the Supplementary are not presented in a consistent manner with those presented in the main manuscript.

Reviewer #2 (Remarks to the Author):

The manuscript titled "Macrophage released ADAMTS1 promotes muscle stem cell activation" describes the role of macrophage derived ADAMTS1 as an activator of muscles satellite cells through impaired NOTCH signaling. The authors provide several compelling lines of evidence supporting their hypothesis. Using single fibre experiments the authors provide evidence that there are more satellite cells present on single fibres treated with recombinant ADAMTS1 as compared to non-treated fibres. They go on to demonstrate that ADAMTS1 RNA and protein expression is upregulated in whole muscle 1 day after muscle injury and demonstrate that ADAMTS1 is co-localized to macrophages. The authors then developed transgenic over-expressers of ADAMTS1 in macrophages and demonstrate that following muscle injury there are more MyoD+ cells and more EdU+ cells (pulse experiments) supporting their hypothesis that ADAMTS1 is an activator of satellite cells. As expected following muscle injury there were more satellite cells present two days following muscle injury and more importantly regeneration as enhanced in 1 month old animals. Surprisingly, regeneration appeared impaired in four month old animals, which was attributed to exhaustion of the satellite cell pool in transgenic animals due to an inability to re-establish quiescence in the ADAMTS1 over-expressers. Additionally, a series of experiments demonstrating that NOTCH is a target of ADAMTS1 ultimately leading to the inhibition of NOTCH signaling and withdrawal of cells from quiescence. The manuscript describes an elegant series of experiments providing compelling evidence that macrophage-derived ADAMTS1 modulates satellite cell activity and could enhance muscle regeneration. They also demonstrate however that ADAMTS1 stimulated satellite cell activity could lead to exhaustion of the satellite cell pool ultimately leading to impaired regeneration/repair. Although overall this is an excellent study there are several shortcomings that need to be addressed.

1. Figure 1 – These experiments are single fibre experiments that show that there are more satellite cell present on recombinant ADAMTS1 treated fibres as compared to untreated control fibres. The image shows two fibres – 1 with more satellite cells (treated) and 1 with less (control).

This experiment does not demonstrate the effectiveness of ADAMTS1 in activating satellite cells. Almost all satellite cells are activated in the process of isolating single fibres and one can find many untreated fibres with many satellite cells clustered similar to the treated fibre shown in the figure. The figure is somewhat misleading as the graphed data shows that the difference between the treated and untreated fibres is not that big (certainly not as big as the authors demonstrate with the fibres selected for the figure). The description of the figure also needs to be tempered as this is not an "activation" experiment per se. There is also no information in the manuscript that this reviewer could find describing the number of fibres analyzed per animal.

2. Line 64-66 – Figure 2d does not show data of co-localization of satellite cells (Pax7) with ADAMTS1. The figure only shows co-localization of ADAMTS1 with macrophages. The statement is misleading.

3. Line 70-73 – It is interesting that ADAMTS1 + macrophages are also present in uninjured muscle. Is there any relationship between the localization of these macrophages and satellite cells in the uninjured condition? Although the frequency of activated satellite cells is very low in the uninjured state it would be interesting to know whether these ADAMTS1+ macrophages are anatomically related to activated satellite cells in the uninjured state.

4. EdU+ myonuclei were quantified in the pulse experiments. Were total myonuclei ever quantified in the transgenic versus the wt animals. This would be interesting to measure since the 1 month old transgenic animals were not different for muscle mass or fibre size, while there was a difference at 4 months of age. What is happening to the additional activated satellite cells if they are not contributing to growth?

5. Figure 4b – the EdU experiments do not tell you about the ability of ADAMTS1 to induce activation. These experiments only tell you that there are more cells dividing. Although a subtle difference, given the nature of the manuscript it may be an important one.

6. Throughout the methods the number of fibers analyzed needs to be specified. The figure legends indicate an N=3 for number of animals but it is not clear how many fibers were included in the analysis.

Reviewer #3 (Remarks to the Author):

In their study, Du et al. investigated the role of ADAMTS1 in satellite cell activation during postnatal growth and regeneration of skeletal muscle. The authors show that ADAMTS1 is secreted by Ly6Cpos macrophages and is responsible for satellite cell activation, that further leads to satellite cell exhaustion at later age. Finally they show specific interaction between ADAMTS1 and Notch, which triggers the inactivation of the later, that activates quiescent satellite cells.

As a whole the study is interesting since, as the authors stated in the introduction, few molecular effectors delivered by macrophages have been involved in the specific steps of myogenesis. While some parts of the study are quite convincing, others present several flaws, the 2 main issues being the effect of ADAMTS1 on satellite activation itself (and not on their expansion) and the relationship driven by ADAMTS1 between inflammatory macrophages and satellite cells, that is lacking.

Major issues:

- Figure 1ab. Single fibers are examined 3 days after isolation (stated in M&M), i.e. after activation, expansion, and differentiation. To precisely analyze the activation step, authors should examine the single fibers few hours after the isolation, e.g. 12 hours. Therefore, they cannot mention (line 56) that the results in Fig1ab "indicate that ADAMTS1 activates satellite cells".
- Figure 2e. Flow cytometry analysis indicates that 100% of Ly6Cpos cells express ADAMTS1, while the ICC pictures show numerous F4/80positive cells that do not express ADAMTS1. This should be clarified. Moreover, antibodies against ADAMTS1 have been used in flow cytometry. Since this is a secreted molecule and since the antibody used is not designed for flow cytometry, the authors should provide the gating strategy, together with the isotypic and relevant control gating.

- The mouse model of overexpression of ADAMTS1 in macrophages is not presented at all. The mice should be described, at least in material and method section (phenotype, fertility, weight, behavior, etc....). Most importantly, the overexpression of ADAMTS1 should be evaluated in macrophages, Fibroadipogenic precursor cells (FAPs) and satellite cells from muscle in both uninjured and injured mice at both 1 and 4 months of age, since the phenotype is different.
- Suppl Fig2 shows similar expression of F4/80 in uninjured muscle. As macrophages have been only poorly in uninjured muscle, and are believed to reside in the fascia (Brigitte et al. Arthritis Rheum. 2010), the authors should precise whether the fascia was included in their analysis, as well as the number of cells recovered, that should be very different in injured and uninjured muscle. Moreover, similar F4/80 staining is observed in both injured and uninjured muscles, which is surprising: in early regenerating muscle, cells ranging from monocytes (F4/80low) to well differentiated macrophages (F4/80high) are present. It is believed that in resting tissues, only resident macrophages (F4/80high) are present. Finally, residing macrophages exhibit, in all tissues examined so far, a Ly6Cneg phenotype. What is the differential expression of ADAMTS1 in Ly6Cpos and Ly6Cneg populations in early, and later time points after injury (and in resting muscle, if there is any Ly6Cpos macrophages identified). Because the authors state that macrophage-derived ADAMTS1 is crucial for satellite cell activation, these providers of ADAMTS1 should be carefully described in both resting and regenerating muscle.
- The EdU experiments are difficult to follow. Were the experiments performed for measuring the EdU+ sat cells identical to those for measuring EdU+ myonuclei? Why is the increase in EdU+ sat cell is two fold while that of myonuclei is 4 fold? At what age the mice have been injected (notably for experiments where authors conclude on post-natal growth)?
 - Fig4ab. Authors injected EdU at day 2 and analyzed the cells at day 3. However, at day 2 after injury, satellite cells are already activated. Thus the results illustrate satellite cell expansion, rather than their activation per se. Injecting EdU at the time of injury, and analyzing at day 1 would be more informative on activation of the cells.
- The demonstration of the impact of macrophage-derived ADAMTS1 on satellite cell behavior is lacking. May be authors could set up cocultures of satellite cells or single fibers with macrophages isolated from WT and from Adamts1 mice to convince about this direct interaction. These experiments should include loss of function experiments (inhibition of ADAMTS1? Use of constitutively active notch satellite cells (If they exist)). Indeed, overexpression of ADAMTS1 in macrophages may trigger over/downregulation of other genes, that lead to the observed effect on satellite cell behavior. Thus the demonstration of a direct functional interaction is required

Minor:

- There are some references in the abstract. This is quite unusual.
- Line 87: experiments are not IHC but ICC.
- Fig4c: histology HE is not good, as compared with the others.
- Fig5ab: in Hey1 analysis, an error appears in the X-axis labeling

Response to Reviewers' comments

We were pleased that the reviewers found our work to be “novel and interesting” and “an excellent study”. The reviewers also had multiple suggestions for improving the study. We greatly appreciate their detailed comments and suggestions and have performed a substantial number of additional experiments as well as revised the text and figures to address these points. We believe the manuscript has been significantly improved by these revisions and thank the reviewers for their thoughtful input. Our point-by-point responses to the reviewers' comments and the changes we have made are detailed below:

Reviewer #1:

Major points

1. Data on the phenotypes of Adamts1-mice is insufficient. Do they grow normally without any apparent defects? It is certain that aP2/FABP4 is highly expressed in macrophages; however, it is also expressed in the adipose tissue, ganglia, cartilage, and vertebrae (at least during embryonic development) (Transgenic Res (2006) 15:647–653). Because ADAMTS1 is a secreted enzyme, overexpression of ADAMTS1 can theoretically have systemic effects. Moreover, as evidenced by various Notch-related gene mutant mice described in past studies, any deterioration in Notch signaling could have a significant impact on cell fate decision, cell growth, etc, in vivo. Given that ADAMTS1 suppresses Notch signaling as the authors claim, it is highly likely that Adamts1-mice exhibit defects that are related to loss of Notch signaling during embryonic development or postnatal growth. Therefore, further characterization of Adamts1 mice is required to validate their conclusions.

Response: We agree with the reviewer that further details about the Adamts1 Tg mice should be provided. The body composition analysis and growth curves for the Adamts1 Tg mice are located in Supplementary Fig. 2 of the revised manuscript. We also agree with the reviewer that aP2 is expressed in other cell types beyond macrophages and, therefore, the transgene might also be expressed in these tissues. In addition to macrophages, we assessed adipose tissue and, in response to the reviewers' points, have now also examined ganglia, cartilage, vertebrae and liver. Our studies show that there is no obvious developmental phenotype in the ganglia, cartilage or vertebrae and, consistent with the embryonic expression of aP2 in these tissues, there is no detectible postnatal expression of ADAMTS1 in these tissues in either wild-type or Adamts1 mice. Please see below figures where we used lung tissue as a positive control to validate that our antibody recognizes ADAMTS1 protein by IHC in wild-type mice (upper panels on IF), as there are cells in this tissue known to express ADAMTS1 (arrows) and since no ADAMTS1 was detected in the other tissues. Adamts1 is overexpressed in adipose tissue in the Adamts1 mice and we found that there is decreased adipose tissue mass in these mice. Of note, these findings further support the importance of the expansion of the muscle tissue to the increased total body mass phenotype observed in these animals (Supplementary Fig. 2). However, we note that the Adamts1 mice are born with the expected Mendelian ratios and we did not observe any additional phenotypes from those discussed above and the muscle phenotype described in detail in the manuscript. A number of possibilities could explain this selectivity but we favor that the tissue-specific expression profile of the transgene is the driving cause and also speculate that there could be context-specificity to the modulation of Notch signaling, as has been observed for a large number of signals. We have revised the paper to include more details on the Adamts1 mouse model used in this study in the Methods section (as suggested by Reviewer 2) and Supplementary Fig. 2. We also expanded the discussion of Notch signaling in the Discussion section (please see below response).

Figure Legend: (Left) Images from H&E stained sections of multiple tissues from Wt and Adamts1 mice. (Right) Upper: IHC of lung tissue from wild-type mice comparing ADAMTS1 antibody staining to IgG control, re-validates the ADAMTS1 antibody for IHC (arrows point to ADAMTS1 positive cells). Lower: IHC of multiple tissues for ADAMTS1.

2. There are notable differences in the phenotypes between Adamts1 mice described in the present study and other types of mutant mice in which Notch-signaling is conditionally abrogated. For example; mutant mice with defective Notch signaling in satellite cells do not usually develop muscle hypertrophy or show increased body weight, at least under unchallenged conditions; depletion of satellite cells does not result in thinner muscle fibers but in an incomplete muscle regeneration which is highlighted by muscle fatty infiltration and accumulation of fibrous tissues. These discrepancies on the phenotype of Adamts1 mice with that of other Notch signaling mutant mice have to be thoroughly examined and discussed.

Response: As the reviewer notes, the phenotype of the Adamts1 Tg mice does not completely phenocopy those mice in which Notch signaling has been conditionally abrogated in satellite cells. However, we would make several key points. First, one way in which the Adamts1 Tg mice do strongly phenocopy the mice in which RBP-Jk is conditionally deleted in satellite cells is the spontaneous activation of satellite cells (Bjornson et al, *Stem Cells*, 2012). This makes sense if Notch signaling is essential to maintain satellite cell quiescence. However, beyond this, we would not expect an exact phenocopy for the following reason: in the RBP-Jk KO satellite cells, Notch signaling is genetically deleted in all of the satellite cell progeny, and it is well known that Notch signaling is important in both the proliferative amplification of progenitors (Conboy and Rando (2002) *Dev Cell*). By contrast, Notch signaling would be expected to be intact, albeit potentially modulated, during the regenerative process in the Adamts1 Tg mice. As such, we would not expect the Adamts1 Tg mice to exhibit this component of the phenotype that occurs when RBP-Jk is genetically ablated in satellite cells.

With regard to body weight and muscle hypertrophy, we did observe this in the Adamts1 Tg mice, but not at one month of age and only by four months of age. Clearly, this could be multifactorial, but we do show compelling data that the Adamts1 Tg mice have increased levels of satellite cell activation (Fig. 3a) and that the increased numbers of activated satellite cells fuse with myofibers (Fig. 3b). These data are consistent with this process contributing to the change in muscle mass, even if this is not the only etiology of the phenotype. Furthermore, it is intriguing that reduced Notch signaling in Pofut1 hypomorphic mice is associated with muscle hypertrophy (i.e. Al Jaam, *Open Bio.*, 2016). Indeed, there are data demonstrating that Notch signaling is both context-dependent and stage-specific (i.e. Bi et al, *eLife*, 2016). We have added text in the Discussion section (lines 220-224) clarifying that additional mechanisms may be contributing.

3. Figure 5i shows that coexpression of Adamts1, but not proteolytically inactive mutant, with Notch1 suppresses the expression of NOTCH1 protein in vitro. Based on this data (and other data showing a decrease in the transcripts for Hey1 and Hes1 in the Adamts1 mice-derived satellite cells), the authors concluded that ADAMTS1 suppresses Notch-signaling through proteolytically processing NOTCH1 and thereby decreasing its expression. This is an interesting hypothesis, which indicates a previously unknown mechanism in regulating the Notch availability on cell surface. However, the data in the present study (Figure 5i) is insufficient to convincingly support the conclusion. Given the impact of this novel hypothesis, highly stringent and extensive data on the processing of NOTCH1 by ADAMTS1 and the consequence of NOTCH1 cleavage must be presented.

Response: We appreciate the reviewer's recognition of the novelty and impact of our findings. In order to examine the effect of ADAMTS1 on NOTCH1 processing in further detail, we overexpressed myc-tagged NOTCH1 to facilitate detection of the processed forms of the protein. We then performed co-culture experiments with cells overexpressing ADAMTS1 and the NOTCH1 ligand DLL1 or negative control cells. We analyzed the processed forms of NOTCH1 using immunoprecipitation of cell lysates from the co-cultures. As shown in the new

Fig. 6j, we found that ADAMTS1 affects the processing of TMIC to NICD, which occurs by cleavage at the extracellular S2 site. We also found that this activity of ADAMTS1 is not rescued by overexpressing NOTCH ligand. The results of these new experiments further corroborate the conclusions and provide additional molecular details on ADAMTS1 effects on NOTCH1.

Minor points

1. *Figure 1: The number of Pax7- MyoD+ cells may also be shown. Please also provide the concentration of ADAMTS1 used in this experiment.*

Response: A new Fig. 1 now also shows Pax7- MyoD+ cells, and the concentration rADAMTS1 that was used (1.4 µg/ml) was added to the figure legend.

2. *Figure 2a: Why does the expression of ADAMTS1 fully subside on day 2, even though there must be a number of macrophages in the injury site? Do these macrophages cease the production of ADAMTS 48 h after the injury? Please clarify this issue or add a discussion on this.*

Response: We agree with the reviewer that it is of interest that the expression of *Adamts1* decreases by day-2 post-injury. To further examine this process, we performed a detailed time-course studying the macrophages that infiltrate the site of muscle injury. Consistent with prior published data by others, we found that Ly6C+ macrophages are in high abundance on day 1 and significantly decline by day 2 (these new data are located in a new Fig 2e). Using RT-qPCR and flow cytometry, we found that the Ly6C+ macrophages express significantly higher levels of *Adamts1* than Ly6C- cells (Fig. 2f,g and Supplementary Fig. 1b) and therefore, this physiological switch in cell types likely explains the change in expression levels observed. While we believe this is the primary mechanism for the observed shift, other mechanisms may also contribute, such as a down-regulation in expression of *Adamts1* in the remaining Ly6C+ cells, but the switch in cell-types at the site of injury is at least a significant contributor to the change.

3. *Figure 2d: F4/80 is a membrane bound protein and is mainly expressed on cell surface. However, Figure 2d shows a positive staining of F4/80 exclusively in the nucleus. This requires an explanation. Furthermore, the result does not necessarily show that satellite cells are negative for ADAMTS1, unless co-staining with a satellite cell marker is presented.*

Response: We apologize if the phase-contrast images we previously provided did not clearly distinguish the nucleus from the outer membrane. We have now replaced the prior images with higher magnification images that demonstrate F4/80 staining outside of the nucleus (new Fig. 2d). Regarding ADAMTS1 levels in satellite cells, we removed the reference to Fig. 2d and added data measuring ADAMTS1 levels in satellite cells as Supplementary Fig. 1a.

4. *As mentioned above, since aP2 is not exclusively expressed in macrophages, bone marrow transfer experiments should be performed to confirm that ADAMTS1 produced by macrophages are responsible for the phenotypes of Adamts1 mice described in the present study.*

Response: We thank the reviewer for the suggestion of an additional very rigorous test of the importance of ADAMTS1 from macrophages for this study. As suggested, we performed bone marrow transplant studies and compared mice that received bone marrow from *Adamts1* mice to mice that received bone marrow from wild-type mice. Importantly, the satellite cell phenotype transplanted with the bone marrow from *Adamts1* Tg mice: Wild-type mice that received bone marrow from the *Adamts1* mice had significantly higher levels of EdU-positive satellite cells post-injury than did wild-type mice that received wild-type bone marrow (Fig. 4b), robustly

confirming the important role of macrophages. These new data are displayed in a new Fig 4b.

5. Figure 4g aims to show a decrease in the population of VCAM+ SCA1- cells in Tg mice compared to Wt mice. However, a significantly more number of cells was analyzed in Wt mice compared to Tg mice. An equal number of cells between Tg and Wt has to be analyzed to present proportional increase or decrease in the number of satellite cells.

Response: We have clarified that an average of $\sim 5 \times 10^5$ cells per mouse were analyzed in both Adamts1 and Wt mice in these studies in the figure legend. In addition, we report the percent of VCAM+ SCA1- in the total live cells, which corrects for any minor differences in absolute numbers between flow cytometry runs, resulting in an accurate comparison across the groups.

6. Figure 4c-e: The peak of the histograms among 1-, 4-, and 8-month old WT mice differs significantly (15-17.5, 12.5-15, and 20-25, respectively). This requires an explanation.

Response: The 8-month old cohorts were evaluated at day-7 post-injury while the other cohorts were analyzed day-5 and this is the likely cause of the difference in the peaks. This was done in case the older mice required a longer period of time for regeneration. We have clarified this in the figure legend. We suspect that the more modest difference between the peaks between the 1-month and 4-month old cohorts represents developmental changes that occur between these time-points.

7. Flow cytometry data in the Supplementary are not presented in a consistent manner with those presented in the main manuscript.

Response: We have revised the format of the flow cytometry data in the Supplementary Data to be consistent with the format used in the main text.

Reviewer #2:

1. Figure 1 – These experiments are single fibre experiments that show that there are more satellite cell present on recombinant ADAMTS1 treated fibres as compared to untreated control fibres. The image shows two fibres – 1 with more satellite cells (treated) and 1 with less (control). This experiment does not demonstrate the effectiveness of ADAMTS1 in activating satellite cells. Almost all satellite cells are activated in the process of isolating single fibres and one can find many untreated fibres with many satellite cells clustered similar to the treated fibre shown in the figure. The figure is somewhat misleading as the graphed data shows that the difference between the treated and untreated fibres is not that big (certainly not as big as the authors demonstrate with the fibres selected for the figure). The description of the figure also needs to be tempered as this is not an “activation” experiment per se. There is also no information in the manuscript that this reviewer could find describing the number of fibres analyzed per animal.

Response: We agree with the reviewer and have repeated these experiments at an earlier time-point and performed IHC to quantify the proportion of MyoD+/Pax7+ and MyoD-/- Pax7+ cells in response to exposure to rADAMTS1, to focus on the effect of ADAMTS1 on the activation of satellite cells (new Fig 1b). We apologize that the reviewer found the images that we previously used in the prior version of Fig. 1 misleading. The intent of the example images was to specify the cells that were being counted in the quantification. However, we can now appreciate that

this was not clear. In order to avoid this confusion, we now show an image of an example of an activated satellite cell (Pax7+/ MyoD+) and a cell that is Pax7+/MyoD-, to demonstrate which cells were quantified on the myofibers (new Fig 1a). For the quantification, each point represents a myofiber, revealing the total number of fibers analyzed (new Fig 1c). This information is also further detailed in the figure legends both for these data and throughout the manuscript to improve clarity.

2. Line 64-66 – Figure 2d does not show data of co-localization of satellite cells (Pax7) with ADAMTS1. The figure only shows co-localization of ADAMTS1 with macrophages. The statement is misleading.

Response: Please see our response to minor point #3 above.

3. Line 70-73 – It is interesting that ADAMTS1 + macrophages are also present in uninjured muscle. Is there any relationship between the localization of these macrophages and satellite cells in the uninjured condition? Although the frequency of activated satellite cells is very low in the uninjured state it would be interesting to know whether these ADAMTS1+ macrophages are anatomically related to activated satellite cells in the uninjured state.

Response: We agree that it is an interesting question if the macrophages in uninjured muscle are located in close proximity to satellite cells. To address this question, we performed IHC along with the RT-qPCR on uninjured muscle tissue. As the reviewer points out, there are very few macrophages (0-2 macrophages per cryosection) in uninjured muscle tissue. However, we were intrigued that these studies showed that, of the few macrophages that are present in uninjured muscle, we frequently found them anatomically located in close proximity to a satellite cell (new Supplementary Fig. 1d). We appreciate the reviewer suggesting this interesting experiment.

4. EdU+ myonuclei were quantified in the pulse experiments. Were total myonuclei ever quantified in the transgenic versus the wt animals. This would be interesting to measure since the 1 month old transgenic animals were not different for muscle mass or fibre size, while there was a difference at 4 months of age. What is happening to the additional activated satellite cells if they are not contributing to growth?

Response: As suggested by the reviewer, we measured the total myonuclei per myofiber and the total myonuclei were increased in the Adamts1 mice compared to the Wt animals (new Supplementary Fig. 2e). We believe these data support that the activated satellite cells are fusing with myofibers (please also see response to Reviewer 1, point 2). These results are also consistent with the data that EdU+ myonuclei were significantly increased in the transgenic mice at 4 months of age. We suspect that it takes time for the process of increasing fiber size and muscle mass to reach quantifiable and significant levels and that this is the reason why we begin to observe this phenotype when the mice reach 4 months of age.

5. Figure 4b – the EdU experiments do not tell you about the ability of ADAMTS1 to induce activation. These experiments only tell you that there are more cells dividing. Although a subtle difference, given the nature of the manuscript it may be an important one.

Response: We agree with the reviewer's comment that proliferation occurs after satellite cell activation and that, while these processes are connected and therefore relevant for our study, the processes occur in distinct phases. We have altered the text (lines: 115-117) to clarify this point.

6. Throughout the methods the number of fibers analyzed needs to be specified. The figure legends indicate an N=3 for number of animals but it is not clear how many fibers were included in the analysis.

Response: We apologize that these method details were not clear: For most experiments, each myofiber is represented by a point on the quantifying graph, thus indicating the total number of myofibers examined. To improve clarity, we have added the myofiber numbers to the figure legends.

Reviewer #3:

Major issues:

- *Figure 1ab. Single fibers are examined 3 days after isolation (stated in M&M), i.e. after activation, expansion, and differentiation. To precisely analyze the activation step, authors should examine the single fibers few hours after the isolation, e.g. 12 hours. Therefore, they cannot mention (line 56) that the results in Fig1ab "indicate that ADAMTS1 activates satellite cells".*

Response: We agree with the reviewer's point and have repeated the experiments at 6 hours. We have revised Fig. 1 accordingly.

- *Figure 2e. Flow cytometry analysis indicates that 100% of Ly6Cpos cells express ADAMTS1, while the ICC pictures show numerous F4/80 positive cells that do not express ADAMTS1. This should be clarified. Moreover, antibodies against ADAMTS1 have been used in flow cytometry. Since this is a secreted molecule and since the antibody used is not designed for flow cytometry, the authors should provide the gating strategy, together with the isotypic and relevant control gating.*

Response: We agree with the reviewer that our studies indicate that nearly 100% of F4/80+ / CD11b+ / Ly6C+ cells express high levels of *Adamts1*. This was confirmed by both flow cytometry and RT-qPCR. We also agree with the reviewer that there are a few cells in the IHC that are F4/80+ but ADAMTS1 low or negative; we believe these cells represent a mixture of cell types including F4/80+ / Ly6C- and F4/80+ / CD11b intermediate or low cells.

We apologize that we did not previously show the isotype control and gating for the ADAMTS1 antibody that were used in the flow cytometry. These data are now included in the Supplementary Data (Supplementary Fig. 1b).

- *The mouse model of overexpression of ADAMTS1 in macrophages is not presented at all. The mice should be described, at least in material and method section (phenotype, fertility, weight, behavior, etc...). Most importantly, the overexpression of ADAMTS1 should be evaluated in macrophages, Fibroadipogenic precursor cells (FAPs) and satellite cells from muscle in both uninjured and injured mice at both 1 and 4 months of age, since the phenotype is different.*

Response: Please see our response to Reviewer 1, Point 1. The mice are born in expected Mendelian ratios and are fertile. They do not have any obvious behavioral abnormalities. We added these and other details about the mouse model to the Methods section ('Animals' subheading) and data on weights and body composition of the mice are presented in Supplementary Figure 2.

In addition, as suggested, we have now conducted additional experiments measuring the expression levels of *Adamts1* in the *Adamts1* Tg mice compared to wild-type littermates in macrophages, satellite cells and FAPs isolated from muscle tissue in both uninjured and injured mice at both 1 and 4 months of age (Supplementary Fig 3a). These new data further corroborate our previous data indicating that *Adamts1* is most highly overexpressed in macrophages.

- *Suppl Fig2 shows similar expression of F4/80 in uninjured muscle. As macrophages have been only poorly in uninjured muscle, and are believed to reside in the fascia (Brigitte et al. Arthritis Rheum. 2010), the authors should precise whether the fascia was included in their analysis, as well as the number of cells recovered, that should be very different in injured and uninjured muscle. Moreover, similar F4/80 staining is observed in both injured and uninjured muscles, which is surprising: in early regenerating muscle, cells ranging from monocytes (F4/80low) to well differentiated macrophages (F4/80high) are present. It is believed that in resting tissues, only resident macrophages (F4/80high) are present. Finally, residing macrophages exhibit, in all tissues examined so far, a Ly6Cneg phenotype. What is the differential expression of ADAMTS1 in Ly6Cpos and Ly6Cneg populations in early, and later time points after injury (and in resting muscle, if there is any Ly6Cpos macrophages identified). Because the authors state that macrophage-derived ADAMTS1 is crucial for satellite cell activation, these providers of ADAMTS1 should be carefully described in both resting and regenerating muscle.*

Response: In our study, the fascia was sacrificed when harvesting the muscle tissues; this information was added to the Methods section (subheading 'Macrophage analysis'). We added the quantification of the absolute numbers of macrophages isolated to Supplementary Fig. 1e. We also added additional data to clarify the transition from F4/80 low to F4/80 high that, in agreement with the reviewer, occurs post-injury: Our data reveal that 5 hours after we injured the mice, most muscle infiltrating macrophages are F4/80 low and this transitions to F4/80 high at post-injury day-1. These data are now shown in Fig. 2e.

As suggested by the reviewer, we quantified the levels of *Adamts1* expression in Ly6C+ compared to Ly6C- macrophages using RT-qPCR. We found that Ly6C+ macrophages express higher levels of *Adamts1* (Fig. 2f). As shown in Fig. 2e, it is important to note that on post-injury day-1, a significant majority of macrophages are Ly6C+ but, by day-4 post injury, the vast majority of macrophages have returned to being Ly6C-. We did find that the small population of Ly6C+ macrophages that were present post-injury day 4 still express higher levels of *Adamts1* than Ly6C- macrophages, as quantified by RT-qPCR. We were unable to examine the expression of *Adamts1* in Ly6C+ cells from uninjured muscle since there are so few of these cells in uninjured tissue.

- *The EdU experiments are difficult to follow. Were the experiments performed for measuring the EDU+ sat cells identical to those for measuring EdU+ myonuclei? Why is the increase in EdU+ sat cell is two fold while that of myonuclei is 4 fold? At what age the mice have been injected (notably for experiments where authors conclude on post-natal growth)?*

Response: We apologize that this experiment was difficult to follow. The experiment for measuring EdU+ satellite cells was performed in the same way as the experiment measuring EdU+ myonuclei. For the experiments where the EdU+ myonuclei were quantified, mice were injected with EdU at 4 months of age. The difference between the number of EdU+ satellite cells and EdU+ myonuclei is most likely caused by the satellite cells undergoing replication after being activated and prior to fusing to the myocytes.

- *Fig4ab. Authors injected EdU at day 2 and analyzed the cells at day 3. However, at day 2 after injury, satellite cells are already activated. Thus the results illustrate satellite cell expansion, rather than their activation per se. Injecting EdU at the time of injury, and analyzing at day 1 would be more informative on activation of the cells.*

Response: We agree with the reviewer and have performed additional experiments, now injecting EdU at the time of injury and analyzing at post-injury day 1. As shown in the new Fig. 4a, there is a significant increase in the number of EdU+ satellite cells in the transgenic mice compared to the wild type controls at this time-point.

- *The demonstration of the impact of macrophage-derived ADAMTS1 on satellite cell behavior is lacking. May be authors could set up cocultures of satellite cells or single fibers with macrophages isolated from WT and from Adamts1 mice to convince about this direct interaction. These experiments should include loss of function experiments (inhibition of ADAMTS1? Use of constitutively active notch satellite cells (If they exist)). Indeed, overexpression of ADAMTS1 in macrophages may trigger over/downregulation of other genes, that lead to the observed effect on satellite cell behavior. Thus the demonstration of a direct functional interaction is required*

Response: We appreciate the importance of demonstrating a direct functional effect of ADAMTS1 on satellite cells and would like to highlight that we performed studies using purified recombinant ADAMTS1 on satellite cells (Fig. 1). We believe that these reductionist experiments robustly demonstrate that ADAMTS1 is acting on satellite cells directly and not by triggering an up- or down-regulation of other genes in macrophages that are secondarily affecting satellite cell behavior. While we attempted to perform loss of function/ knockdown experiments, several different constructs that we tested were unable to significantly decrease ADAMTS1 levels. However, we performed two additional lines of experiments that both corroborate our data supporting the role of macrophage-derived Adamts1 on satellite cell behavior: (1) We co-cultured macrophages with satellite cells and monitored satellite cell activity (Supplementary Fig. 3b). (2) We performed bone marrow transplantation studies (described above) that confirmed that the satellite cell phenotype is transplanted with the bone marrow. Therefore, using multiple orthogonal approaches, we have confirmed that macrophage-derived ADAMTS1 modulates satellite cell behavior.

Minor:

- *There are some references in the abstract. This is quite unusual.*

Response: We removed these references.

- *Line 87: experiments are not IHC but ICC.*

Response: We corrected this error.

- *Fig4c: histology HE is not good, as compared with the others.*

Response: We replaced this image.

- *Fig5ab: in Hey1 analysis, an error appears in the X-axis labeling*

Response: We corrected this error.

We again thank all the reviewers for their very helpful comments and suggestions and believe the manuscript has been significantly improved by their input.

REVIEWERS' COMMENTS:

Reviewer #1 (Remarks to the Author):

The revised manuscript has somewhat improved; however, there still remain issues that make the present study less convincing. The present study deals with two major subjects; 1) the potential function of macrophage-derived ADAMTS1 in promoting SC activity, and 2) the potential function of ADAMTS1 in suppressing Notch activity. Each of these subjects are of interest and can potentially serve as a basis for the future studies. In this regard, the data in each subject need to be highly stringent. Unfortunately, there are issues that compromise the integrity of the present study,

The potential role of ADAMTS1 in regulating the Notch signaling is shown only in Figure 6. Given the importance and impact of this finding, more extensive and stronger data are required. Most critically, there is no data on the potential mechanism by which ADAMTS1 suppresses the Notch signaling. It is also not clear to what extent the regulation of the Notch signaling by ADAMTS1 is physiologically relevant in vivo or whether this is a universal mechanism in regulating the activity of Notch signaling. Analysis of the Tg mouse model in the present study does not necessarily tell whether ADAMTS secreted from macrophages is actually involved in the regulation of satellite cell activation under physiological conditions. Furthermore, the results of the animal studies are somewhat ambiguous to draw any solid conclusions and raise many questions.

Reviewer #2 (Remarks to the Author):

Thank you for your thorough response to my comments. The manuscript is significantly improved over the first iteration.

Reviewer #3 (Remarks to the Author):

The authors adequately answered to all the concerns I raised. Thank you

Response to reviewers

We thank the Editor and all the reviewers for their careful consideration of our revised manuscript and we were pleased that both reviewers #2 and #3 found our revised manuscript improved, that we had “*thorough responses*” to the comments and that we “*answered all of the concerns*”. However, Reviewer #1 had additional concerns. We detail our response to Reviewer #1’s concerns below and with additional textual changes in the manuscript.

Reviewer #1 comments:

The revised manuscript has somewhat improved; however, there still remain issues that make the present study less convincing. The present study deals with two major subjects; 1) the potential function of macrophage-derived ADAMTS1 in promoting SC activity, and 2) the potential function of ADAMTS1 in suppressing Notch activity. Each of these subjects are of interest and can potentially serve as a basis for the future studies. In this regard, the data in each subject need to be highly stringent. Unfortunately, there are issues that compromise the integrity of the present study,

The potential role of ADAMTS1 in regulating the Notch signaling is shown only in Figure 6. Given the importance and impact of this finding, more extensive and stronger data are required. Most critically, there is no data on the potential mechanism by which ADAMTS1 suppresses the Notch signaling. It is also not clear to what extent the regulation of the Notch signaling by ADAMTS1 is physiologically relevant in vivo or whether this is a universal mechanism in regulating the activity of Notch signaling. Analysis of the Tg mouse model in the present study does not necessarily tell whether ADAMTS secreted from macrophages is actually involved in the regulation of satellite cell activation under physiological conditions. Furthermore, the results of the animal studies are somewhat ambiguous to draw any solid conclusions and raise many questions.

Response to Reviewer #1 comments:

First, we would like to point out that our studies revealing the recruitment of ADAMTS1 secreting macrophages to the sites of muscle injuries were performed in wild-type mice, supporting that our conclusions are physiologically relevant. We disagree that “there is no data on the potential mechanism by which ADAMTS1 suppresses the Notch signaling”- we demonstrated that ADAMTS1 interacts with NOTCH1 in co-IP experiments, that the effect of ADMATS1 on NOTCH1 is dependent on an active metalloproteinase and that ADAMTS1 alters NOTCH1 processing. We agree with the reviewer that our studies uncovered a previously unknown mechanism to regulate satellite cell activity and Notch signaling. We also agree that further studies examining in greater detail this molecular mechanism and the broader implication for Notch signaling in other contexts are of great interest. However, we contend that the fact that we have revealed a novel pathway with potentially broad implications for wide ranging future studies in a variety of contexts underscores, rather than mitigates, the novelty and impact of our findings. As both other reviewers concur, our results provide robust support for our conclusion that Adamts1 has an important and previously unrecognized role in regulating satellite cell activation with implications for muscle regeneration, the focus of this study. To further respond to reviewer’s concerns, we have made additional textual changes in the Discussion section highlighting the caveats that future studies could reveal more details on the mechanisms by which Adamts1 regulates Notch signaling and additional physiological implications for muscle and other tissues.

Reviewer #2 Comments:

Thank you for your thorough response to my comments. The manuscript is significantly improved over the first iteration.

Reviewer #3 Comments:

The authors adequately answered to all the concerns I raised. Thank you